# Long-Term Safety of Omalizumab in Children with Asthma and/or Chronic Spontaneous Urticaria: A 4-Year Prospective Study in Real Life

**DOI:** 10.3390/jpm13071068

**Published:** 2023-06-29

**Authors:** Francesca Galletta, Lucia Caminiti, Cecilia Lugarà, Simone Foti Randazzese, Paolo Barraco, Federica D’Amico, Pierangela Irrera, Giuseppe Crisafulli, Sara Manti

**Affiliations:** 1Pediatric Unit, Department of Human Pathology in Adult and Developmental Age “Gaetano Barresi”, University of Messina, 98124 Messina, Italy; francygall.92@gmail.com (F.G.); lucycaminiti@yahoo.it (L.C.); cecilialug@gmail.com (C.L.); paolo.barraco@outlook.it (P.B.); federicadamico93@gmail.com (F.D.); crisafulli.giuseppe@unime.it (G.C.); saramanti@hotmail.it (S.M.); 2Department of Clinical and Experimental Medicine, University of Messina, 98124 Messina, Italy; pierangela.irrera@unime.it

**Keywords:** asthma, children, chronic spontaneous urticaria, omalizumab, real-life study, safety

## Abstract

**Background:** Insufficient data are available on the long-term “real-life” safety profile of omalizumab in children. This study evaluated the long-term safety of omalizumab in a pediatric cohort with severe asthma or chronic spontaneous urticaria (CSU). **Methods:** A monocentric, prospective study evaluated the long-term safety of omalizumab in patients aged 6–18 years. Each patient completed the standardized MedDRA questionnaire to identify adverse events (AEs). **Results:** In total, 23 patients, median age 15 (14–18) years, affected by severe asthma (60.8%) or CSU (39.2%), treated with omalizumab for 2 (1–4) years were enrolled. The most common AEs belong to the system organ class (SOC) of general disorders and administration-site conditions (37.17%). Skin and subcutaneous tissue problems represent the second most frequently reported AEs (24.35%). Central nervous system and musculoskeletal disorders were quite frequent (15.38% and 8.97%, respectively). Other adverse events were tachycardia (5.12%), vertigo and abdominal pain (2.60% and 3.86%, respectively), and dry eye (1.3%). Only one patient reported herpes virus infection during treatment (1.3%). No cases of anaphylaxis, hemopathies, uronephropathies, respiratory, psychiatric, hepatobiliary, or oncological pathologies were reported. **Conclusions:** Long-term “real-life” treatment with omalizumab in children appears well tolerated. Its safety and efficacy profile makes omalizumab an excellent alternative in severe asthma and CSU in children.

## 1. Introduction

Omalizumab is a recombinant monoclonal antibody that binds to the Fcε portion of immunoglobulin E (IgE) [1]. The binding between allergens and allergen-specific IgE (sIgE) on the surface of mast cells and basophils leads to the rapid release of inflammatory mediators, including histamine. This process results in clinical manifestations of type 1 hypersensitivity and the promotion of adhesion and infiltration of circulating inflammatory cells. According to this underlining mechanism of action of IgE antibodies in allergy, it is evident that anti-IgE therapy could represent an effective treatment for several allergic diseases [1]. Specifically, omalizumab decreases levels of circulating IgE, downregulating FceRI expression on inflammatory cells, leading to a reduction in the presentation to T cells and inhibition of the Th2-mediated allergic pathway [1]. Consequently, omalizumab decreases pro-inflammatory mediator release, thus reducing allergic inflammation [1]. All the effects justify its clinical efficacy; therefore, the development of omalizumab, about 40 years ago, had a significant impact on the management of allergic diseases. Nowadays, omalizumab is approved by Food and Drug Administration (FDA) for treating moderate to severe persistent asthma in patients aged >6 years, chronic spontaneous urticaria (CSU) in patients aged >12 years, and nasal polyps in individuals older than 18 years of age [2]. In addition, several studies are ongoing involving omalizumab as an add-on to oral immunotherapy (OIT) or as a monotherapy for food allergy [3].

Omalizumab is recommended to be administered as a subcutaneous injection. For asthma therapy, the dose and frequency of dosing are based on total serum IgE level and body weight, while for CSU, the dose is the same for each patient [4]. Since its approval, several studies have demonstrated the efficacy of omalizumab in allergic diseases, leading to a reduction in asthma exacerbations, with a decrease in the use of inhaled corticosteroids (ICS), improvement and/or resolution of urticaria, and improvement of quality of life [5,6,7,8]. Regarding the safety profile, several randomized clinical trials (RCTs) and post-marketing surveillance studies demonstrated that omalizumab is generally well tolerated with a very low frequency and severity of adverse events (AEs) [9,10]. However, safety data of long-term treatment with omalizumab are insufficient in children. Most studies investigated safety for about 2 follow-up years, and only a few report data for more extended periods [11,12]. Our study aimed to evaluate, in real life, the long-term safety profile of pediatric patients affected by asthma and/or CSU treated with omalizumab.

## 2. Materials and Methods

### 2.1. Type of Study

A prospective, single-center, observational, real-life study was conducted from January 2019 to January 2023 at the Paediatric Unit of AOU G. Martino, University of Messina, Messina, Italy. The long-term safety profile of omalizumab in pediatric patients was defined as the number and type (mild, moderate, severe) of adverse events (AEs) in patients who have been receiving omalizumab treatment for at least 30.73 ± 29.5 months. The AEs were registered during the regular follow-up visit; moreover, for each AE reported, the physician recorded a detailed description, including time to onset and recovery, seriousness, outcome, and codifying the AEs according to the Medical Dictionary for Regulatory Activities (MedDRA^®^) Preferred Term (PT), and System Organ Class (SOC) levels.

The local research ethics committee approved the study (code 99-22, 16 September 2022).

### 2.2. Patients

Pediatric patients affected by severe asthma and/or CSU treated with omalizumab were recruited. The following inclusion criteria were adopted: age older than 6 years old; confirmed diagnosis of severe asthma in agreement with the current guidelines, and receiving therapy with one biologic drug [13]. Patients with a confirmed diagnosis of CSU in accordance with the current guidelines were also included in the study [14]. CSU was diagnosed in patients with a history and characteristic skin lesions (wheals or/and angioedema) lasting more than six weeks and with a poor response to antihistamines, anti-leukotrienes, and systemic steroids for short-term therapy. Patients lost to follow-up or with other concomitant chronic diseases or with poor compliance to treatment were excluded from the study. All patients included in the trial and the parents of patients were appropriately informed and signed an informed consent agreement. The study was conducted according to Good Clinical Practice and in compliance with the Declaration of Helsinki with successive amendments [15].

### 2.3. Dosing

For patients who have severe asthma, the dose applied for omalizumab was 150–600 mg subcutaneously every two or four weeks, depending on baseline serum total IgE level measured before the start of the treatment and body weight [16]. The administered dose of omalizumab to patients with CSU was 300 mg by subcutaneous injection every four weeks [17].

### 2.4. Statistical Methods

The continuous variables of the sample were categorized using descriptive statistics.

The data were analyzed using Statistical Package for Social Sciences (SPSS) version 22.0. Data were considered statistically significant with a *p*-value < 0.05. The continuous variables were expressed as mean ± standard deviation (SD) or median value, as appropriate. The ordinary variables were expressed as percentages. Fisher’s exact test or the Pearson Chi-squared test (Pearson coefficient of correlation) for qualitative variables and the paired t-test for continuous variables were used. The Naranjo Adverse Probability Scale (NAPS) [18] was assumed to estimate the association between AEs and drug treatment. According to this algorithm, we classified the AEs into doubtful (0), possible (1–4), probable (5–8), and certain (>8). This evaluation was performed for each AE reported by the patients or children’s parents. Post hoc power analysis was performed using the post hoc Power Calculator (clincalc.com (accessed on 10 June 2023)). Since data on a control group were not available, we calculated the post hoc power comparing AEs that occurred in our sample with drug-related AEs reported in a larger population of patients treated with omalizumab.

## 3. Results

In total, 23 patients (11 females and 12 males) with a diagnosis of severe asthma and/or CSU were enrolled. The clinical and demographic features of this population are shown in Table 1. Among them, 14 patients are affected by severe asthma (6 females and 8 males), and 9 patients are affected by CSU (7 females and 2 males). The mean age of the population is 14.18 ± 16.67 years. The mean duration of treatment with omalizumab is 30.73 ± 29.5 months. The most frequent comorbidity presented by patients included rhinoconjunctivitis (13 patients), followed by obesity (5 patients); comorbidities are shown in Table 1.

### Prevalence and Type of AEs

The AEs are reported in Table 2. Of the 23 enrolled patients, 16 patients (69.5%) reported one or more AEs. Our patients reported a total number of 77 AEs. The principal AEs of the SOC were “general disorders and administration site conditions” (37.17%), and overall, the principal AE of the preferred team (PT) was asthenia (11 patients, 14.10% of overall reactions), followed by administration-site reaction (8 patients, 10.25%). Additional AEs were reported in the following SOCs: “skin and subcutaneous tissue disorders” (19 AEs, 24.35%), “nervous system disorders”(12 AEs, 15.38%), “musculoskeletal and connective tissue disorders” (7 AEs, 9%), “cardiac disorders” (4 AEs, 5.19%), “gastrointestinal disorders” (3 AEs, 3.9%), “ear and labyrinth disorders” (2 AEs, 2,6%), “infection and infestations” (1 AEs, 1.3%), and “eye disorders” (1 AEs, 1.3%). The most common AEs, by PT, following omalizumab administration were asthenia (11 patients, 14,28%), pruritus (9 patients, 11.68%), drowsiness (6 patients, 7.79%), administration-site reaction (8 patients, 10.25%), headache (5 patients, 6.49%), urticaria (5 patients, 6.49%), and tachycardia (4 patients, 5.19%). One patient developed a herpes virus infection that required therapy and temporarily suspended omalizumab. No severe AEs occurred in this group of patients during the therapy with omalizumab. Immune system disorders, hepatobiliary disorders, psychiatric disorders, renal and urinary disorders, blood and lymphatic system disorders, and cancer were not reported. None of the enrolled patients discontinued omalizumab for AEs.

## 4. Discussion

This 4-year prospective observational study showed that long-term therapy with omalizumab in children is safe and well tolerated in a real-life setting. Globally, among our cohort, 69.5% experienced AEs. A similar incidence rate was shown in other real-life studies and post-marketing surveillance studies [19,20]. On the contrary, in RCTs, the incidence rate was higher (≈90%) [21,22]. A recent multicenter, short-term, single-arm study was conducted enrolling 1528 patients, including adults, adolescents, and pediatric patients (≥6 years old). Among the overall population, 23.6% and 4.5% of patients reported AEs and severe AEs, respectively. Among a total of 191 pediatric patients, 14.1% and 1.6% of patients reported AEs and severe AEs, respectively [23]. In the overall population, the severity of AEs was mostly mild (13.5%) and moderate (7.9%). Only 2.2% of the patients had severe AEs.

The most common AEs reported belong to the SOC of general disorders and administration-site conditions (37.17%). These data were in line with a recent study, in which this category was the most frequent in a pediatric population (3.9%) [20]. According to Nakamura et al., among 128 patients, the most common AE was hyperpyrexia (2.4%) [20]. However, in our study population, only two patients experienced hyperpyrexia, and the correlation with omalizumab administration was not clear. Similar to other studies [24,25,26], asthenia was the most frequent AE among the same SOC, followed by administration-site local reactions. Nevertheless, it was not necessary to discontinue the treatment, as this kind of reaction did not represent a problem for the patient, even in the long term. Moreover, according to drug manufacturers, in adults and adolescents, injection-site reactions occurred in 45% of the cases compared with 43% in placebo-treated patients [27]. Hyperidrosis and flushing were not commonly detected, as they were reported in two cases and one case, respectively. In a study conducted in the Czech Republic, including patients aged >12 years, affected by severe allergic asthma, only 1 patient experienced flushing and hyperidrosis associated with headache, tremors, and dry mouth [28]. Two patients reported peripheral edema. To the best of our knowledge, no RCTs showed this AE associated with using omalizumab. One of these patients was affected by CSU; thus, it is reasonable to hypothesize that the underlying disease could elicit the clinical picture. Three of our patients experienced chest pain. This AE was described in the literature as a symptom consequent to other complications, such as inflammation of blood vessels or heart disease [27]. Narukonda et al. [29] described a case of pulmonary vein thrombosis, onset with chest pain and dyspnea, in an adult male with severe persistent asthma under omalizumab for 2 years. Therefore, this correlation should be confirmed by other studies. Among our cohort, no patient reported cardiovascular disease, although the US Food and Drug Administration (FDA) paid attention to the incidence of cardiovascular AEs in adults and adolescents [30]. This was based on the EXCELS study that showed an increased risk of cardiovascular and cerebrovascular events, such as CV death, myocardial infarction, ischemic stroke, transient ischemic attack, and unstable angina, in patients receiving omalizumab aged >12 years [31]. Four patients reported tachycardia after the administration. This isolated symptom is rarely reported in the literature [32]. Cildag S. described a case of an adult male who had to discontinue omalizumab therapy because it was the trigger of atrial fibrillation [33]. Therefore, we cannot exclude that in our patients, it was a consequence of an emotional state. The second SOC most frequently reported was skin and subcutaneous tissue disorders. Among these, pruritus was the most-reported AE (9 cases), associated or not with urticaria (5 cases), rash (3 cases), and erythema (2 cases). In a double-blind placebo-controlled study conducted by Milgrom et al. [34], in children with severe allergic asthma, skin rashes and urticaria were mild, with a similar prevalence in both groups. Notably, four out of five patients who reported urticaria suffered from CSU and started omalizumab for this reason. Di Bona et al. [12] described only one patient with urticaria and angioedema who experienced the occurrence of similar episodes also before starting omalizumab. Moreover, this AE was considered rare by the manufacturers; thus, it is possible that it occurs mainly in patients who are already affected by this disease. Nervous system disorders resulted quite frequently. Among these, drowsiness and headache were the most common AEs. Only one patient reported a confused state. Headache and drowsiness were frequent AEs reported by RCTs in children, with an incidence rate of 13.8–36% [27,34,35]. The ANCHOR study was a multicenter, observational, retrospective cohort study, involving 25 pediatric allergy and pulmonology patients, conducted between 2006 and 2018. Among 484 patients analyzed, 21 experienced AEs, and headache was the most frequently reported (1.7%) [36]. Similar results were derived from a recent multicenter study in which 41.7% of children affected by CSU and treated with omalizumab reported headache [26]. No patient experienced peripheral nerve disorders. A short observational study showed no peripheral neuropathy in any of the patients included but did show altered amplitude, latency, and velocity values of the peripheral nerves [37].

Vertigo was reported by two patients (2.60%). This AE is rarely described in children, but more frequent in adults. Bhutani et al. [38] reported only 1 patient (1%) with vertigo and drowsiness in a study population of 99 adults with allergic asthma. Similar data (1.16%) were reported in another recent retrospective study on adults with CSU [39]. A study analyzed individual case safety reports from the Uppsala Monitoring Centre VigiBase of biologics for asthma up to 29 December 2019. In 32,618 reports, 17 AEs were detected as signals of omalizumab-related ear and labyrinth disorders [40]. This report showed that ear disorders are rare, but still should be considered.

In our cohort, 3.9% reported abdominal pain without other PTs belonging to the same SOC, such as vomiting, nausea, or dyspepsia. In clinical trials, upper abdominal pain and nausea were common in patients aged <12 years old [24,41,42]. However, in the ANCHOR study, the percentage of abdominal pain reported by pediatric patients was very low (0.2%) [36]. According to manufacturers, arthralgia, limb, and back pain could occur, with an onset 1 to 5 days after the first or subsequent injections of omalizumab, although rarely [27,42]. Among our study population, 9.09% of patients reported musculoskeletal disorders, mainly back pain, but no one stopped the treatment. On the contrary, Di Bona et al. [12] reported that arthralgia was the leading cause of discontinuation in their cohort. One patient experienced dry eye. This is not an AE related to omalizumab administration; thus, more studies are necessary to confirm these data. We did not report any case of parasitic infection, although blocking IgE may theoretically increase this risk. Our long-term finding was also confirmed by a specific study conducted in Brazil [43]. However, one patient developed a herpes virus infection that required therapy and the temporary suspension of omalizumab, without recurring again. This AE was not reported in RCTs or observational studies. Di Bona et al. [12] described a case of relapsing herpes labialis related to omalizumab administration, suggesting a possible role of omalizumab in the immune control of specific viral infections. No patients reported respiratory disorders. This is a significant result for patients suffering from asthma and CSU. In some RCTs, asthma exacerbation was considered an AE related to omalizumab administration [42,44]. Our asthmatic patients reported just an improvement in their disease. In a previously cited study, the most commonly reported AEs by PT in the overall population were upper respiratory tract infection, asthma, and nasopharyngitis, with incidence rates of 3.5%, 3.0%, and 2.5%, respectively. Specifically, in the pediatric population, the most common AEs reported were upper respiratory tract infection and nasopharyngitis, with incidence rates of 4.7% and 1.6%, respectively [23]. Regarding the risk of malignancy, no patient developed any cancer. This is in line with the EXCELS study, which demonstrated that the risk of malignancy was similar in omalizumab and non-omalizumab users [45]. However, an analysis of the VigiBase pharmacovigilance database identified a significantly higher risk of malignancies in the adult population, with a strong correlation to breast and lung cancer [46]. These data need further investigation. Another significant result of our study was that no patients experienced allergic reactions during the treatment years. Data from RCTs and post-marketing surveillance studies showed that hypersensitivity reactions to omalizumab are not common, and anaphylaxis is very rare, occurring in about 0.09% of patients [47]. Female sex represents a risk factor for developing anaphylaxis [48]. Safety data from real-life short-term observational studies are in line with RCTs [49]. These results were also confirmed by other long-term real-life studies, both in adults and children [12,50]. However, a retrospective study using data from the US FDA Adverse Event Reporting System (FAERS) database identified a total of 2006 cases of anaphylaxis associated with biological drugs from January 2004 to September 2020 in young and middle-aged adults. Omalizumab was associated with the largest number of anaphylaxis cases compared to other biologics, but it has been on the market for nearly 20 years, longer than the others [51].

Particular attention should also be paid to excipients. Following the administration of omalizumab, a case of angioedema with positive skin prick tests for polysorbate was recently reported [52]. Interestingly, no statistically significant differences were found between patients treated with high and low doses of omalizumab. Thus, we could deduce that the incidence of adverse events did not depend on omalizumab dosage, even over the long term. Similar data are reported in other studies, in which it was demonstrated that the incidence of side effects during high-dose treatment was not elevated compared to standard treatment [25,26]. Important data emerged: no AEs reported by our patients led to the discontinuation of the treatment. Previous studies reported a very low discontinuation rate due to AEs [12,53]. However, this result was in contrast with other real-world data, in which pediatric patients with asthma had a higher rate of discontinuation due to AEs, with fatigue being the most frequently reported [24,54]. The strength of our study was its real-life nature, compared with RCTs and post-marketing surveillance studies. The RCTs are designed to test if the selected treatment is working, but they do not evaluate if the treatment works in real life. Aiming to limit any factor that could potentially influence final results, RCTs are designed in compliance with a rigid and strict protocol; however, the latter is not the same in all RCTs, and thus, “within-study” and “between-study” heterogeneity are extensively reported among different trials [55]. Moreover, the RCTs investigate an experimental treatment in a selected group of patients who must respect the study’s inclusion criteria [46]. Any protocol modification is foreseen for patients who do not strictly meet the inclusion criteria or unforeseen events [55]. Thus, it can happen that, due to several reasons (e.g., age, disease severity, comorbidities, use of concomitant medications, etc.), a cluster of patients, from which it could be possible to extract potential and interesting data, must be excluded from the study, resulting in a gap in knowledge of the investigated treatment in real practice. Although on the one hand, the “one-size-fits-all” approach can provide the opportunity to apply a treatment for a large population, on the other hand, it does not work for everyone. A treatment defined as “effective” in an ideal clinical setting cannot give specific quantifiable answers under individual cases in a routine clinical setting. Tailoring the healthcare approach and treatment to meet the specific needs of each patient is urgently needed, as only a “sartorial” approach can consider an individual’s unique molecular, lifestyle, and clinical information. We strongly believe that real-world studies, using data collected in everyday clinical settings, can hold the promise of providing real data to maximize the applicability and generalizability of an intervention.

## 5. Conclusions

Long-term “real-life” treatment with omalizumab in children appears well tolerated. Long-term exposure to omalizumab is not related to increased incidence of AEs. The good safety profile, associated with an already proven efficacy, makes omalizumab an excellent alternative in the management of severe asthma and CSU in children. However, further long-term “real-life” prospective studies are needed to confirm these data.

## Figures and Tables

**Table 1 jpm-13-01068-t001:** Demographic and clinical findings of the enrolled population.

Age (Years, Median)	15 (14–18)
M:F	12:11
Race	Caucasian
Num. of pts. affected by	
• Asthma	14
• CSU	9
Indications for starting treatment	Severe asthma CSU
Dose for pts. affected by	
• Asthma	The dose applied for omalizumab was 150–375 mg subcutaneously every 2 or 4 weeks, depending on baseline serum total IgE level measured before the start of the treatment, and body weight
• CSU	300 mg every 4 weeks
Duration treatment (years, median)	2 (1–4)
Comorbidities	Rhinoconjunctivitis (num. pts. 13) Obesity (num. pts. 5) Food allergies (num. pts. 2) Type 1 diabetes (num. pts. 1) Hashimoto’s Thyroiditis (num. pts. 1) Drug allergy (num. pts. 1) Hypereosinophilia (num. pts. 1) Selective IgM deficiency (num. pts. 1)

Num.: number; pts.: patients; CSU: chronic spontaneous urticaria.

**Table 2 jpm-13-01068-t002:** MedDRA-compliant description of adverse events (AEs).

SOC—General Disorders and Administration-Site Conditions	29
PT—Pyrexia	2
PT1—Asthenia	11
PT2—Hot flush	1
PT3—Pallor	0
PT4—Administration-site reactions	8
PT5—Peripheral edema	2
PT6—Chest pain	3
PT7—Hyperhidrosis	2
SOC—Cardiac disorders	4
PT—Tachycardia	4
PT2—Myocardial infarction	0
SOC—Skin and subcutaneous tissue disorders	19
PT—Folliculitis	0
PT1—Rash	3
PT2—Pruritus	9
PT3—Erythema	2
PT4—Psoriasis	0
PT5—Parapsoriasis	0
PT6—Urticaria	5
PT7—Pityriasis	0
PT8—Purpura	0
SOC—Ear and labyrinth disorders	2
PT—Vertigo	2
SOC—Nervous system disorders	12
PT—Headache	5
PT1—Drowsiness	6
PT2—Confused state	1
PT3—Paraesthesia	0
SOC—Infections and infestations	1
PT—Cytomegalovirus infection	0
PT1—Herpes virus infection	1
PT2—Pyelonephritis	0
PT3—Subcutaneous abscess	0
PT4—Abscess oral	0
PT5—Cystitis	0
PT6—Clostridia infection	0
PT7—Tinea versicolor	0
SOC—Respiratory, thoracic, and mediastinal disorders	0
PT—Pneumonia	0
PT1—Asthma	0
PT2—Nasopharyngitis	0
PT3—Tonsillitis	0
PT4—Dyspnea	0
PT5—Laryngospasm	0
PT6—Pulmonary embolism	0
SOC—Blood and lymphatic system disorders	0
PT—Leukocytosis	0
PT1—Leukopenia	0
PT2—Anemia	0
PT3—Neutropenia	0
PT4—Lymphadenopathy	0
PT5—Lymphadenitis	0
PT6—White blood cell disorder	0
SOC—Gastrointestinal disorders	3
PT—Vomiting	0
PT1—Gingivitis	0
PT2—Nausea	0
PT3—Tooth loss	0
PT4—Abdominal pain	3
PT5—Dyspepsia	0
SOC—Immune system disorders	0
PT—Allergic reaction	0
PT1—Anaphylactoid reaction	0
SOC—Renal and urinary disorders	0
PT—Nephrolithiasis	0
SOC—Benign, malignant, and unspecified neoplasms	0
PT—Colorectal cancer	0
PT1—Skin cancer	0
PT2—Lung adenocarcinoma	0
SOC—Musculoskeletal and connective tissue disorders	7
PT—Myalgia	0
PT2—Limb discomfort	3
PT3—Back pain	3
PT4—Arthralgia	1
SOC—Eye disorders	1
PT—Dry eye	1
PT1—Blurred vision	0
SOC—Psychiatric disorders	0
PT—Libido decreased	0
SOC—Hepatobiliary disorders	0
PT—Cholelithiasis	0

## Data Availability

The data are not publicly available due to privacy.

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
