# Peer review of "Long-Term Safety of Omalizumab in Children with Asthma and/or Chronic Spontaneous Urticaria: A 4-Year Prospective Study in Real Life"

_jpm, 2023, doi:10.3390/jpm13071068_

Round 1

Reviewer 1 Report

Dear Authors,

Thanks for valuable manuscript.

Tables need major revision. Table 1 not repeat text of result and just mentioned demographic like median age and comorbids and others. Table 2 need just positive points and description of abbreviations.

Igm deficiency is diagnosed with whole exom in patient and also hyper eosinophilic syndrome or patient just had high eosinophil?

just minor native edit

Author Response

Reviewer: 1

Dear Authors,

Thanks for valuable manuscript.

  1. Tables need major revision. Table 1 does not repeat text of result and just mentioned demographic like median age and comorbidities and others. Table 2 need just positive points and description of abbreviations.
  • As you suggested, we made the necessary changes in the two tables in the paper.
  1. IgM deficiency is diagnosed with Whole Exome in patient and also hyper-eosinophilic syndrome, or patient just had high eosinophils?
  • IgM deficiency was diagnosed by blood tests showing low levels of IgM, while hyper-eosinophilic syndrome was suspected and diagnosed because of high levels of eosinophils.

Reviewer 2 Report

Manuscript reference: jpm-2473050

Type of article: article

Title:  Long-term safety of omalizumab in children with asthma 2 and/or chronic spontaneous urticaria: A 4-year prospective 3 study in real-life.

 Studies that analyze the side effects of a drug such as omalizumab are very important and needed, especially those involving long-term periods, nevertheless, the studied group is very small since 23 patients are involved.

My main concern is that author expressed the data in peculiar way, for instance; mean  age 14.18 (±16.67) years. This would mean that there was a patient in the group who was, for example, 30 years old. Similarly, treatment with omalizumab for 30.73 (±29.5) months would mean that someone was treated for a month, another even 60 months. I think these data should be expressed as medians with IQR (interquartile range), which would more clearly show the age distribution and duration of omalizumab treatment among patients.

Minor: Page 2, line 91; after  “According to this algorithm”, remove were from “we were classified”.

Minor editing of English language required.

Author Response

Reviewer: 2

  1. Studies that analyze the side effects of a drug such as omalizumab are very important and needed, especially those involving long-term periods, nevertheless, the studied group is very small since 23 patients are involved. My main concern is that author expressed the data in peculiar way, for instance; mean age 14.18 (±16.67) years. This would mean that there was a patient in the group who was, for example, 30 years old. Similarly, treatment with omalizumab for 30.73 (±29.5) months would mean that someone was treated for a month, another even 60 months. I think these data should be expressed as medians with IQR (interquartile range), which would more clearly show the age distribution and duration of omalizumab treatment among patients.
  • As you suggested, we made the necessary changes in the table in the paper.
  1. Minor: Page 2, line 91; after “According to this algorithm”, remove were from “we were classified”.
  • Thanks for the comment. As you suggested, we made the necessary changes in the text.